# ATM regulation of IL-8 links oxidative stress to cancer cell migration and invasion

Wei-Ta Chen[1], Nancy D Ebelt[1,2], Travis H Stracker[3], Blerta Xhemalce[1], Carla L Van Den Berg[1,2], Kyle M Miller[1]*

[1]Department of Molecular Biosciences, Institute for Cellular and Molecular Biology, University of Texas at Austin, Austin, United States; [2]Division of Pharmacology and Toxicology, College of Pharmacy, Dell Pediatric Research Institute, University of Texas at Austin, Austin, United States; [3]Oncology Programme, Institute for Research in Biomedicine, Barcelona, Spain

**Abstract** Ataxia-telangiectasia mutated (ATM) protein kinase regulates the DNA damage response (DDR) and is associated with cancer suppression. Here we report a cancer-promoting role for ATM. ATM depletion in metastatic cancer cells reduced cell migration and invasion. Transcription analyses identified a gene network, including the chemokine IL-8, regulated by ATM. *IL-8* expression required ATM and was regulated by oxidative stress. IL-8 was validated as an ATM target by its ability to rescue cell migration and invasion defects in ATM-depleted cells. Finally, ATM-depletion in human breast cancer cells reduced lung tumors in a mouse xenograft model and clinical data validated IL-8 in lung metastasis. These findings provide insights into how ATM activation by oxidative stress regulates *IL-8* to sustain cell migration and invasion in cancer cells to promote metastatic potential. Thus, in addition to well-established roles in tumor suppression, these findings identify a role for ATM in tumor progression.

*For correspondence: kyle.miller@austin.utexas.edu

Competing interests: The authors declare that no competing interests exist.

## Introduction

Cancer treatments rely heavily on DNA damaging agents, including radiation and chemotherapeutics, to eliminate cancer cells and decrease tumor burden (*Begg et al., 2011*; *Lord and Ashworth, 2012*; *Cheung-Ong et al., 2013*). These cancer therapies activate complex signaling networks, termed the DNA damage response (DDR), that detect and signal the presence of DNA damage to promote cell cycle arrest and DNA repair (*Jackson and Bartek, 2009*; *Ciccia and Elledge, 2010*). The apical protein kinase ataxia-telangiectasia mutated (ATM) initiates a large signaling cascade in response to DNA double-strand breaks (DSBs) by phosphorylating many key proteins, including the tumor suppressor p53, to orchestrate the DDR (*Shiloh and Ziv, 2013*; *Stracker et al., 2013*). The DDR regulates cell fate decision pathways including apoptosis, senescence and differentiation. Many of these pathways depend on ATM and/or p53 to enforce tumor suppressive anti-cancer barriers to limit proliferation in response to, and in the presence of, DNA damage (*Norbury and Zhivotovsky, 2004*; *Bartkova et al., 2005*; *Gorgoulis et al., 2005*; *Bartkova et al., 2006*; *Di Micco et al., 2006*; *Vousden and Lane, 2007*; *Sherman et al., 2011*; *Roos and Kaina, 2013*). However, cancer cells exhibit genome instability (*Hanahan and Weinberg, 2011*) and often contain endogenous oxidative and replicative damage that can promote genetic alterations to drive malignant transformation (*Klaunig et al., 2010*; *Hills and Diffley, 2014*). Cancer cells evolve a defective DDR to allow cell proliferation in the presence of DNA damage, furthering genome instability and cancer progression. Defects in the DDR can also profoundly influence DNA damage-dependent therapies both positively and negatively (*Bouwman and Jonkers, 2012*). Thus, DDRs can influence both cancer promoting and suppressing mechanisms. This concept is perhaps best exemplified by p53, which is normally activated by stress

**eLife digest** Damaged DNA threatens the normal activity of living cells, so cells use a number of mechanisms to ensure that this damage is repaired. When DNA is damaged, an enzyme called ATM activates several other proteins that ultimately lead to the DNA being repaired. Problems with detecting and repairing damaged DNA have been linked to cancer. Thus, these pathways, including the activity of ATM, were previously thought to only be involved in cancer suppression.

Now, Chen et al. report a new cancer-promoting role for ATM. The experiments reveal that reducing the amount of ATM in cancer cells actually made them less able to migrate and less invasive. Likewise, human breast cancer cells in which the levels of ATM had been depleted formed fewer lung tumors than normal breast cancer cells when they were transplated into mice.

Oxidative stress—a build-up of harmful chemicals inside cells—is a signature feature of cancer cells and is known to be another signal that activates ATM. Chen et al. found that activating ATM through oxidative stress, but not by DNA damage, encouraged the cancer cells to migrate and become invasive. Further analysis of cellular responses following ATM activation by oxidative stress revealed that this enzyme regulates the production of a small protein called interleukin-8. This protein is an important pro-inflammatory molecule that has been implicated in cancer, in particular, in helping cancer cells to migrate to other tissues. When interleukin-8 was added to ATM-depleted cancer cells, it rescued their defects in spreading and invasiveness, thereby providing strong evidence that interleukin-8 is a biologically important target of ATM.

Clinical data confirmed that breast cancer cells that had also spread to the patient's lungs often produced high levels of interleukin-8. Together, these findings suggest that ATM could be a potential target for anti-cancer therapies, as inhibiting this enzyme would inhibit interleukin-8, and in turn slow the progression and spread of cancer.

signals and promotes tumor suppressor pathways (*Bieging et al., 2014*). However, the tumor-associated gain-of-function p53 mutations that are most commonly observed in human cancers exhibit several tumor-promoting functions (*Muller and Vousden, 2013*). Thus, tumor suppressive or promoting activities from the same protein can be dictated by many factors including mutations, cell type, and/or disease context.

ATM regulates complex signaling networks that are involved in many biological processes in addition to DSB signaling and repair (*Stracker et al., 2013*). Aside from increased tumorigenesis, ATM-deficiency results in altered metabolism, aberrant immune and inflammatory responses and increased levels of reactive oxygen species (ROS, *Schneider et al., 2006*; *Alexander et al., 2010*; *Freund et al., 2011*; *Kulinski et al., 2012*; *Valentin-Vega et al., 2012*). Several of the pathological outcomes in ATM deficient mice have been linked to ROS and many of these pathologies can be reversed by the addition of antioxidants, highlighting the important role of ATM in regulating redox-homeostasis (*Ito et al., 2004*; *Reliene and Schiestl, 2006, 2007*; *Reliene et al., 2008*; *Freund et al., 2011*; *Okuno et al., 2012*; *D'Souza et al., 2013*). In addition, ATM can be directly activated by ROS, independently from DSB signaling, and has been implicated in mitochondrial quality control, potentially through an ability to localize to mitochondria (*Guo et al., 2010*; *Valentin-Vega et al., 2012*). The substrates of ATM following ROS mediated activation, and how distinct they are from those modified following DNA damage, remains unknown.

The NF-κB family of transcription factors are similarly activated by multiple stimuli that include DNA damage and ROS. NF-κB signaling regulates inflammation and is involved in cancer where it has been associated, like p53, with both anti and pro-tumorigenic processes (*Ben-Neriah and Karin, 2011*; *Oeckinghaus et al., 2011*). NF-κB signaling can be regulated by ATM through the phosphorylation of NEMO, which aids in transmitting nuclear ATM signaling to activate NF-κB in the cytoplasm (*Wu et al., 2006*). Upon activation, NF-κB translocates to the nucleus where it regulates gene expression and influences cell survival pathways. Understanding the crosstalk between the NF-κB and ATM signaling in complex pathologies including inflammation and cancer remains an area of active investigation.

Cancer cells acquire several capabilities in addition to uncontrolled proliferation during the multistep process of tumorigenesis, including cell migration and invasion that promote tumor

metastasis. (*Friedl and Alexander, 2011*; *Hanahan and Weinberg, 2011*; *Valastyan and Weinberg, 2011*). Metastasis occurs when a tumor spreads by means of cell migration and invasion to a secondary site, a process linked with the majority of cancer deaths (*Hanahan and Weinberg, 2011*). Several pathways suppress this dangerous process as part of tumor suppressive mechanisms (*Mehlen and Puisieux, 2006*). Although the role of DNA damage and the DDR in this process has not been extensively explored experimentally, increased genomic instability due to the acquisition of DDR defects has been proposed to play a role in the acquisition of these traits based on human patient data (*Halazonetis et al., 2008*).

Here, we studied the contribution of the DDR kinase, ATM, in metastatic programs operating in cancer cells. We found that ATM supports migration and invasion, cellular processes intimately linked with metastasis. Our data is consistent with endogenous oxidative stress triggering these responses independently from DSB signaling. Gene expression analysis identified an ATM and mutant p53 transcriptional program that contained pro-migration and invasion genes, including interleukin-8 (IL-8). Addition of IL-8 rescued defects in cell migration and invasion observed in ATM-depleted cells thus validating this target. Consistent with the biological relevance of these findings, ATM promoted tumor formation in a xenograft human breast cancer cell model. Based on the data, we propose a non-canonical role for ATM in supporting pro-tumorigenic behavior of cancer cells.

## Results

### ATM promotes cell migration and invasion independent of DNA DSBs

Given the prevalence and success of DNA-damage based cancer treatments, we sought to determine whether DNA damage and the DDR could inhibit pro-metastatic processes as part of its well-established role as an anti-cancer barrier. To this end, we first determined the effects of DNA damaging agents on cell migration, a process intimately linked with metastasis. DNA damage, including ionizing radiation (IR), topoisomerase II poisons etoposide (Et) and doxorubicin (Dox), had little effect on cell migration in the highly metastatic human breast cancer cell line MDA-MB-231 as measured by wound healing assays (*Figure 1A*, quantified in *Figure 1B*). Induction of the DNA damage marker, phosphorylated histone variant H2AX (γH2AX), confirmed DNA DSBs formations (*Figure 1B*). These treatments arrested the cell cycle and inhibited proliferation (*Figure 1C,D*). Thus, cell migration is independent from DSB induction by DNA damaging agents, as well as cell cycle and growth arrest.

Defects in DDR genes and DNA damage signaling affect cellular responses to chemotherapeutic drugs in some cancer cells (*Bouwman and Jonkers, 2012*). As ATM mediates many initial DNA damage signaling events and regulates multiple cellular responses to DNA damage, we tested its role in cell migration. Strikingly, ATM depletion reduced cell migration in the presence or absence of IR-induced damage (*Figure 1E*, quantified in *Figure 1F*). Increased DDR signaling as evidenced by phosphorylated H2AX, p53 and KAP1 confirmed DNA damage by IR (*Figure 1F*). We independently verified the requirement of ATM for cell migration and invasion using an xCelligence system that measures cell migration and invasion, as well as proliferation in real-time and in the same cell populations which provides standardized experimental conditions. Reduced invasion and migration in ATM-depleted cells was independent from proliferation as control and ATM-depleted cells grew similarly (*Figure 1G*). Experiments with two additional independent ATM siRNAs confirmed these observations thus ruling out possible off-target effects by siRNAs (*Figure 1—figure supplement 1*). Thus, ATM promotes cell migration and invasion in MDA-MB-231 cells in the absence of exogenous DNA damage.

### ATM and p53 operate to promote cell migration and invasion

ATM is activated by at least two distinct mechanisms (*Shiloh and Ziv, 2013*). DSBs activate ATM through interactions with the MRE11-RAD50-NBS1 (MRN) complex. ATM phosphorylates many targets, including the effector kinase CHK2, which collectively orchestrate the DDR (*Bakkenist and Kastan, 2003*; *Lee and Paull, 2005*; *Matsuoka et al., 2007*; *Shiloh and Ziv, 2013*). ATM is also activated by oxidative stress independently of DSBs, MRN or CHK2 (*Guo et al., 2010*), although the downstream pathways reliant on this ATM activation are unclear. In contrast to ATM inhibition, depletion of CHK2, MRE11 or NBS1 did not reduce cell migration, suggesting ATM-mediated cell migration occurred independently of DSB-dependent ATM activation (*Figure 2A,B*; depletion analysis

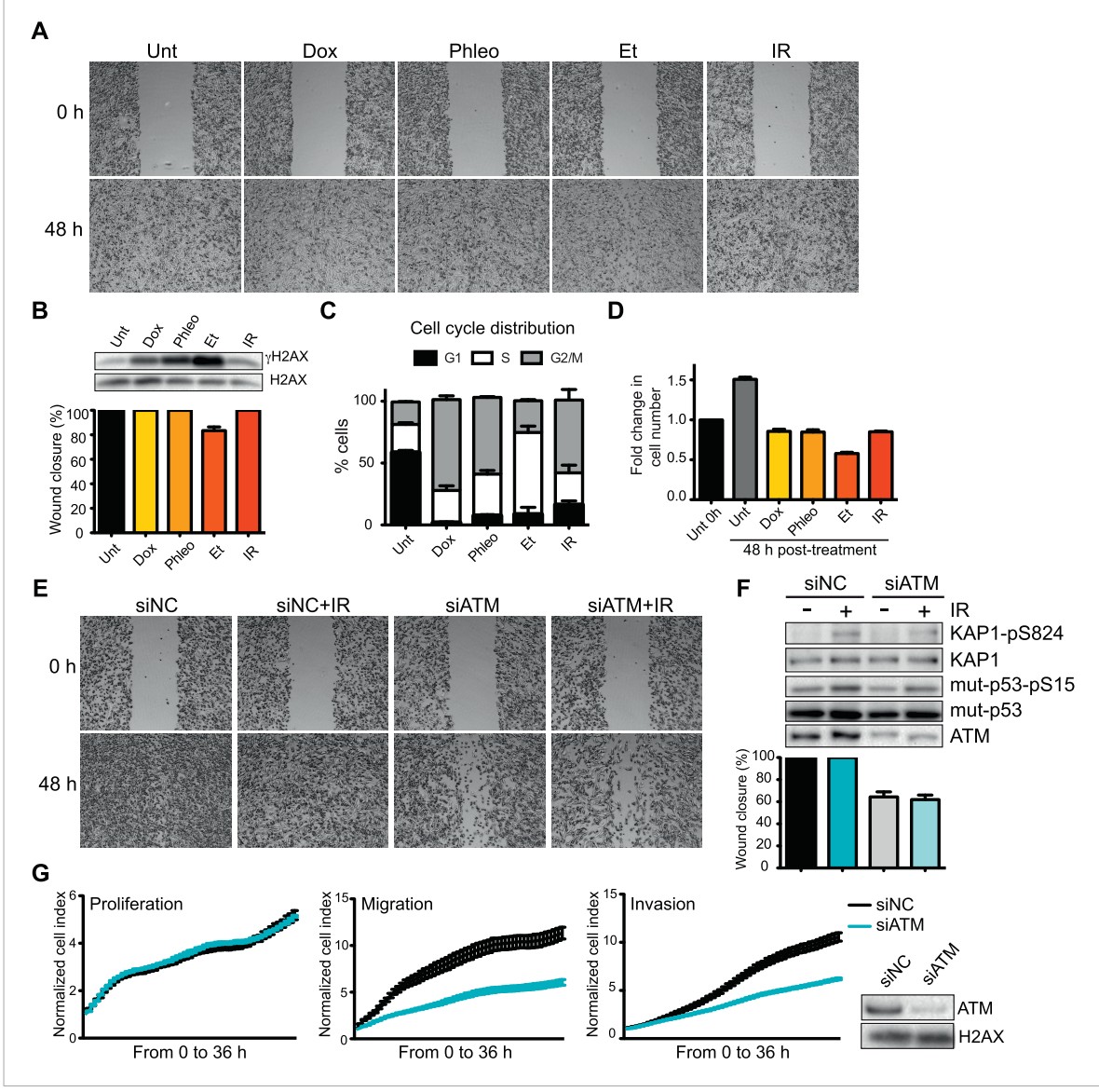

**Figure 1**. Ataxia-telangiectasia mutated (ATM) is required for cell migration and invasion in MDA-MB-231 cells. (**A**) Wound-healing assays of MDA-MB-231 cells untreated (Unt) or treated with doxorubicin (Dox, 100 nM), phleomycin (Phleo, 90 µg/ml), etoposide (Et, 40 µM) or ionizing radiation (IR, 20 Gy). Drug treatments were for 48 hr. IR treatment was performed at time 0. All samples were analyzed post-48 hr from wound induction. Images were acquired at 0 and 48 hr. Representative images from three independent experiments are shown. (**B**) Verification of DNA damage induction and quantification of wound healing from (**A**). Top: Western blot analysis of samples from (**A**) with the DNA damage marker γH2AX. H2AX is a loading control. Bottom: Quantification of wound healing experiments from (**A**). (**C**) Cell cycle analysis of cells treated in panel **A** by flow cytometry. Cells were treated as in (**A**) and analyzed by FACS 48 hr post-treatment. (**D**) Proliferation of cells treated as in (**A**). After 48 hr, cells were trypsinized, counted and normalized to untreated cells at 0 hr. (**E**) ATM promotes cell migration in the absence of induced double-strand breaks (DSBs). Wound-healing assays were performed in siRNA-treated MDA-MB-231 human breast cancercells with siNon-coding (siNC) or siATM siRNAs. (**F**) Verification of DNA damage induction and quantification of wound healing from (**E**). Top: Western blot analysis of samples from (**E**) with the ATM markers KAP1-pS824 and mut-p53-pS15. Unmodified proteins are loading controls and ATM controls siRNA depletion. Bottom: Quantification of wound healing experiments from (**E**). (**G**) ATM depletion impairs cell migration and invasion, but not proliferation. Left panel: siNC or siATM cells were analyzed with xCELLigence Real-time cell analyzer (RTCA) to measure proliferation, migration and invasion in parallel and real-time. Experiments performed as detailed in 'Materials and methods'. Right panel: ATM depletion 96 hr post-transfection. (mean ± s.e.m., n = 3).

The following figure supplement is available for figure 1:

**Figure supplement 1**. Knockdown of ATM impairs cell migration and invasion.

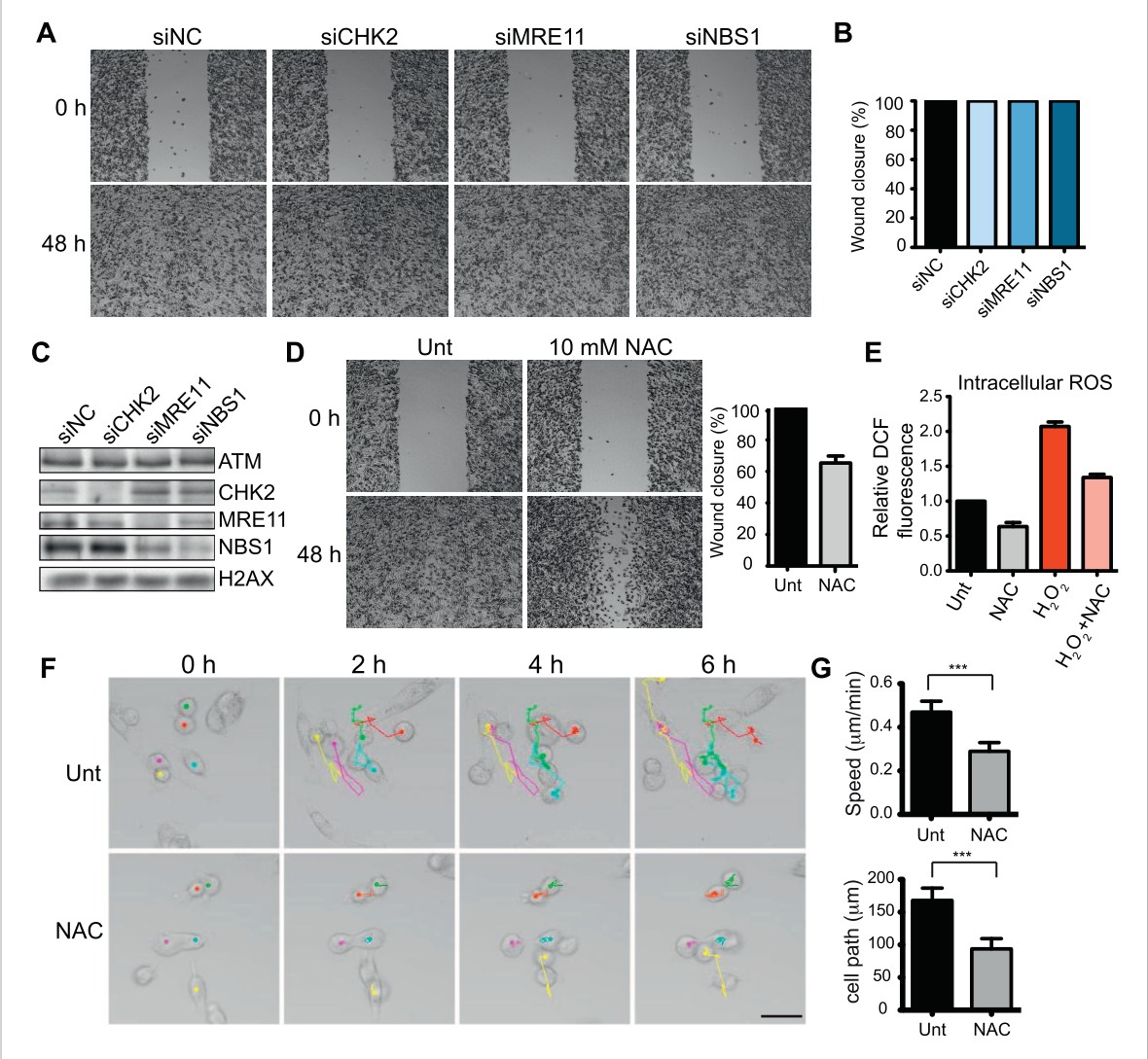

**Figure 2**. ATM promotes cell migration and invasion independently of DNA DSB signaling in MDA-MB-231 cells. (**A**) DSB signaling is not involved in cell migration. Experiments were performed as in *Figure 1A* with the indicated siRNAs. (**B** and **C**) Quantification of wound healing (**B**) and siRNA depletions (**C**) in (**A**). (**D**) Reactive oxygen species (ROS) inhibitor N-acetylcysteine (NAC) reduces cell migration. Right panel: quantification of wound healing. (**E**) NAC treatment reduces endogenous ROS. Cells treated with 10 mM NAC were analyzed using an intracellular ROS detector as detailed in 'Materials and methods'. 4 mM $H_2O_2$ treatment serves as a positive control. (**F**) Live-imaging analysis of cells treated with 10 mM NAC or left untreated. Images were acquired every 15 min for 6 hr and cell were tracked using ImageJ. Colored dots and lines represent individual cell paths. Scale bar, 37.5 µm. (**G**) Quantification of individual cell speed (µm/min) and cell path (µm) from (**F**). Cell parameters were quantified in ImageJ and represent mean data from >100 cells. Error bars = SD. *** p-value <0.0001, unpaired two-tailed t-test.

The following figure supplement is available for figure 2:

**Figure supplement 1**. Inhibition of oxidative stress reduces cell migration and invasion.

shown in *Figure 1C*). Thus, we hypothesized that ATM could promote migration through oxidative stress. In support of this idea, chemical inhibition of oxidative stress in MDA-MB-231 cells using the reducing agent N-acetylcysteine (NAC) reduced cell migration similarly as depletion of ATM (*Figure 2D* and *Figure 2—figure supplement 1*). We also showed that NAC treatment inhibits intracellular oxidative stress (*Figure 2E*). To rule out any influence of proliferation on cell migration in our analyses, we performed live-cell imaging to track the migration of individual cells. We found reduced cell motility in cells treated with NAC compared with control cells (*Figure 2F,G* and *Videos 1, 2*). These data suggest that ATM, independent from DSB signaling, promotes cell migration.

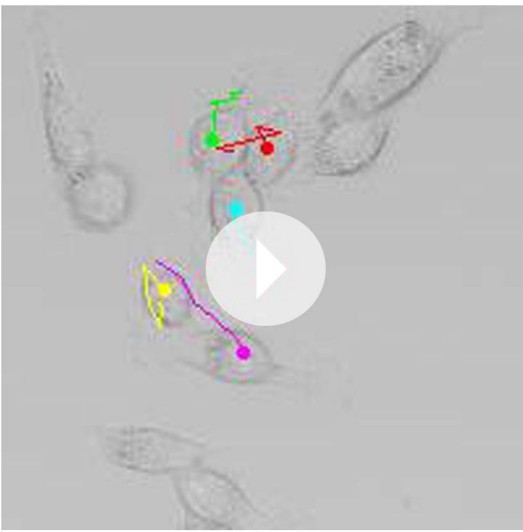

**Video 1.** Live cell imaging and tracking of untreated MDA-MB-231 cells for *Figure 2*. Images were taken every 15 min for 6 hr and tracking was performed in ImageJ. Still images and quantifications are provided in *Figure 2E,F*.

While DSBs trigger ATM-dependent phosphorylation of more than 1000 proteins (*Matsuoka et al., 2007*; *Bennetzen et al., 2010*; *Bensimon et al., 2010*), very few targets of ATM in response to other types of damage, including oxidative stress, are known (*Guo et al., 2010*). However, p53 is phosphorylated by ATM after oxidative stress and mutant p53 is involved in TGFβ-dependent cell migration in MDA-MB-231 cells (*Adorno et al., 2009*; *Guo et al., 2010*). We therefore speculated that mutant p53 could act in concert with ATM to promote cell migration. To test this hypothesis, we depleted ATM, mutant p53 or both and analyzed cell migration. As expected, inhibition of mutant p53 resulted in reduced cell migration (*Figure 3A*, quantified in *Figure 3B*). Interestingly, co-depletion of ATM and mutant p53 resulted in a similar reduction of cell migration as either single gene knockdown alone (*Figure 3A*, quantified in *Figure 3B*). We confirmed and extended these results using real-time, simultaneous analyses of proliferation, migration and invasion. Although mutant p53 depletion mildly reduced proliferation, co-depletion of ATM resulted in an epistatic reduction of cell migration and invasion (*Figure 3C*). Live-cell imaging of migrating cells revealed reduced speed and migration path length in ATM and mutant p53 depleted cells (*Figure 3D,E* and *Videos 3–5*). These results further corroborate the role of ATM and mutant p53 in promoting cell migration, independently from cell proliferation. These results are consistent with our analyses of DNA damaging agents, which inhibit proliferation without altering cell migration (*Figure 1A–D*). We repeated these experiments in the highly metastatic BT-549 breast cancer cells that have mutant p53. Consistent with results from MDA-MB-231 cells, depletion of ATM reduced cell migration by ∼ 60% in BT-549 (*Figure 3—figure supplement 1A*). ATM and/or mutant p53-depleted BT-549 cells exhibited similar levels of migration (*Figure 3—figure supplement 1B*), which is in accord with data obtained in MDA-MB-231 cells (*Figure 3A–C*). Taken together, these results from multiple human cancer cell lines suggest that ATM and mutant p53 are required for the cell migration and invasion phenotypes observed in these highly invasive cancer cell lines.

## ATM regulates interleukin-8 (IL-8)

We next investigated the molecular mechanism by which ATM promotes cell migration and invasion. The identification of an epistatic relationship between ATM and mutant p53 in promoting cell migration pointed towards a common molecular pathway. Given the role of mutant p53 in transcriptional regulation, we hypothesized that ATM could similarly regulate the expression of genes involved in cell migration (*Riley et al., 2008*; *Shiloh and Ziv, 2013*).

**Video 2.** Live cell imaging and tracking of MDA-MB-231 cells treated with 10 mM NAC for *Figure 2*. Images were taken every 15 min for 6 hr and tracking was performed in ImageJ. Still images and quantifications are provided in *Figure 2E,F*.

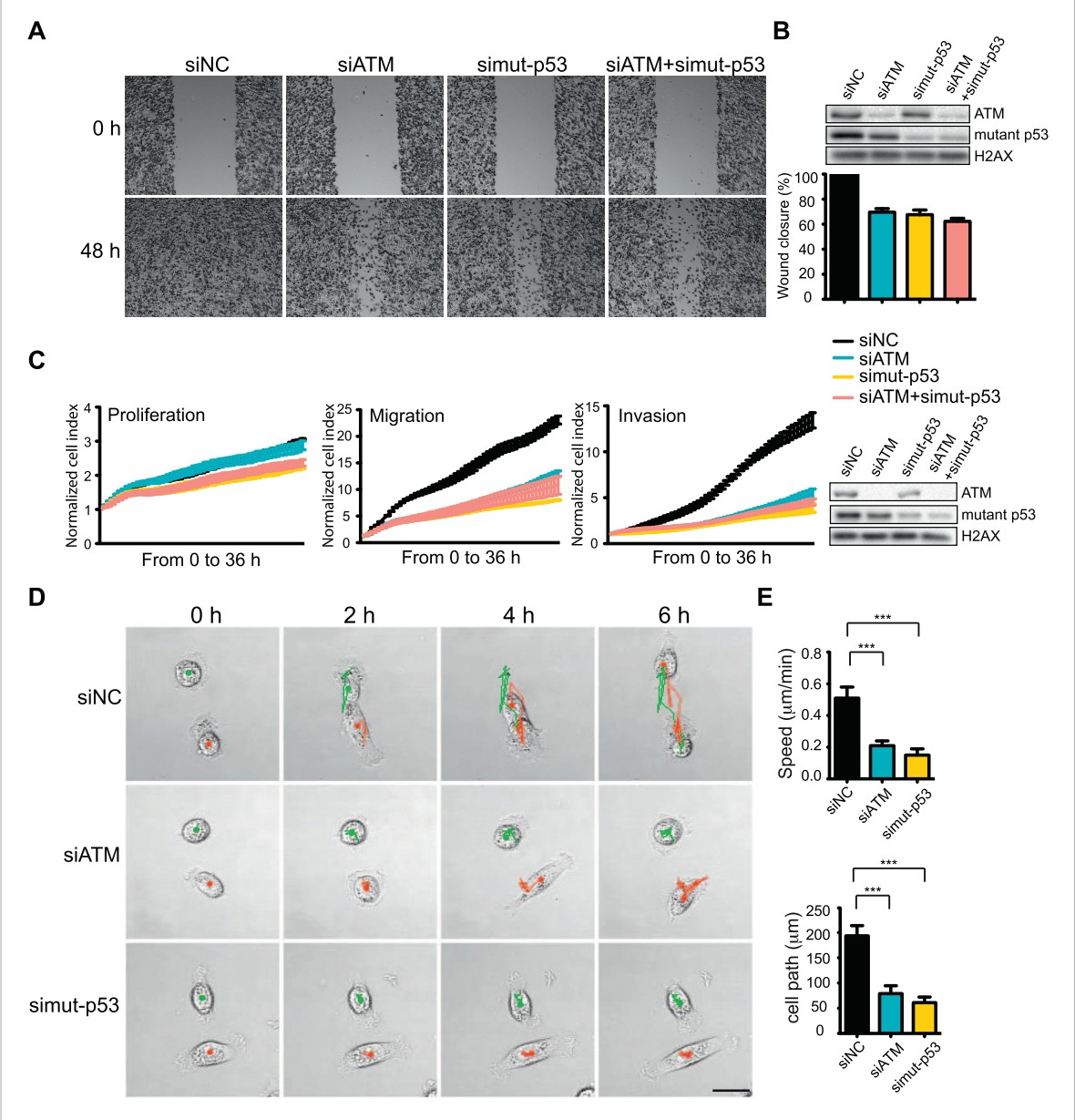

**Figure 3**. ATM-mutant p53 axis of the DNA damage response (DDR) promotes cell migration and invasion in MDA-MB-231 cells. (**A**) ATM or mutant p53 depletion, as well as co-depletion, impairs cell motility similarly. Wound-healing assays were performed with the indicated siRNAs as in *Figure 2A*. (**B**) siRNA depletions and quantification of wound healing for (**A**). (**C**) Real-time analysis of cell dynamics in siATM, simutant-p53 and co-depleted cells. Experiments performed as in *Figure 1G* with indicated siRNAs. Right: ATM and mutant p53 levels in cell samples. (**D**) Live cell imaging of cell migration defects in ATM and mutant p53 depleted MDA-MB-231 cells. Experiments were performed and analyzed as in *Figure 2F*. Scale bar, 20 μm. (**E**) Quantification of individual cell speed (μm/min) and cell path (μm) from (**D**). Cell parameters were quantified as in *Figure 2G*. Error bars = SD. *** p-value <0.0001, unpaired two-tailed t-test.

The following figure supplement is available for figure 3:

**Figure supplement 1**. Analysis of ATM and mutant p53 functions in human breast cancer cells.

Therefore, we performed a microarray-based comparative gene expression analysis in control, ATM and mutant p53 siRNA-treated cells. Out of the hundreds of genes in ATM-depleted cells whose mRNA levels were differentially regulated more than 1.5-fold compared to control siNon-coding (siNC) cells, only ~40 genes showed equivalent regulation between siATM and si mutant p53 cells

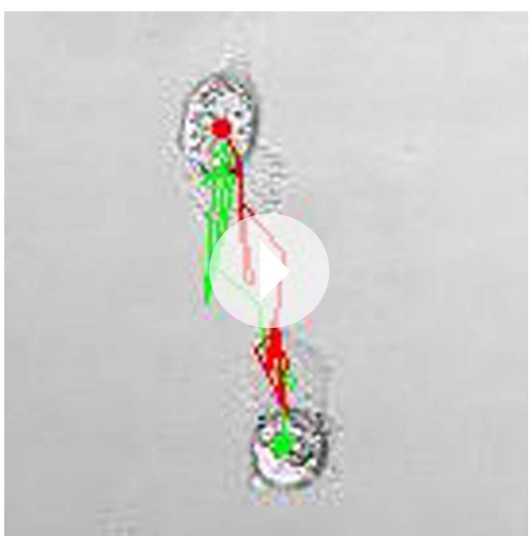

**Video 3.** Live cell imaging and tracking of siNC MDA-MB-231 cells for *Figure 2*. Images were taken every 15 min for 6 hr and tracking was performed in ImageJ. Still images and quantifications are provided in *Figure 3D,E*.

(*Figure 4A* and *Supplementary file 1A,B*). We observed comparable numbers of co-regulated genes that were either up-regulated or down-regulated similarly in both siATM and si mutant p53 cells (*Figure 4A* and *Supplementary file 1C*). Gene ontology (GO) analysis indicated that genes involved in the response to wound healing, including cell migration genes, were in the top 10 GO categories for both ATM and mutant p53 depleted samples (*Figure 4—figure supplement 1*). GO analysis of genes co-regulated by ATM and mutant p53 identified almost exclusively pathways involving cell migration (*Figure 4—figure supplement 1C* and *Supplementary file 1C*) in agreement with our data showing reduced migration in ATM or mutant p53 deficient cells. Collectively these data support a role for mutant p53 and ATM in the co-regulation of a gene network regulating cell migration under these conditions.

We focused our analysis on the down-regulated genes identified in both ATM and mutant p53 data sets since GO analysis only identified the wound healing pathway in this gene set (*Figure 4—figure supplement 1*). Interestingly, the inflammatory cytokine interleukin-8 (IL-8) was the most down-regulated gene in both siATM and si mutant p53 cells (*Figure 4A*). We confirmed *IL-8* down-regulation in mutant p53-containing cell lines MDA-MB-231 and BT-549 upon ATM depletion (*Figure 4B,C*). Conversely, depletion of ATM in cancer cell lines containing WT p53 resulted in increased IL-8 mRNA levels (*Figure 4—figure supplement 2A,B*). These results suggest that ATM promotes IL-8 levels in the context of mutant p53.

*IL-8* is upregulated in several cancers, including breast cancer, where it mediates several cancer promoting pathways including cell migration (*Campbell et al., 2013*; *Singh et al., 2013*). The *IL-8* promoter contains many transcription factor binding sites, including NF-κB, which regulates IL-8 expression and is linked to the DDR through ATM activation by DSBs (*Mukaida et al., 1990*; *Biton and Ashkenazi, 2011*; *McCool and Miya-moto, 2012*). We confirmed *IL-8* promoter regulation by NF-κB as ~90% of IL-8 promoter activity was lost by mutating the NF-κB binding site (mut IL-8, *Figure 4D*). Interestingly, depletion of ATM or mutant p53 reduced *IL-8* promoter activity similarly as mut *IL-8*, showing ATM regulation of *IL-8* occurs at the transcriptional level (*Figure 4D*). As expected, we observed that depletion of NF-κB p65, a subunit of NF-κB dimer, or NEMO abrogated *IL-8* expression in MDA-MB-231 (*Figure 4E*, *Freund et al., 2004*). Both ATM and p53 are known to be required for NF-κB localization and activation in the nucleus upon various stimuli including cellular stress (*Wuerzberger-Davis et al., 2007*; *Hoesel and Schmid, 2013*). To determine whether NF-κB function required ATM or mutant p53 in our cell system, we investigated the nuclear localization of the NF-κB subunit p65 in MDA-MB-231 cells

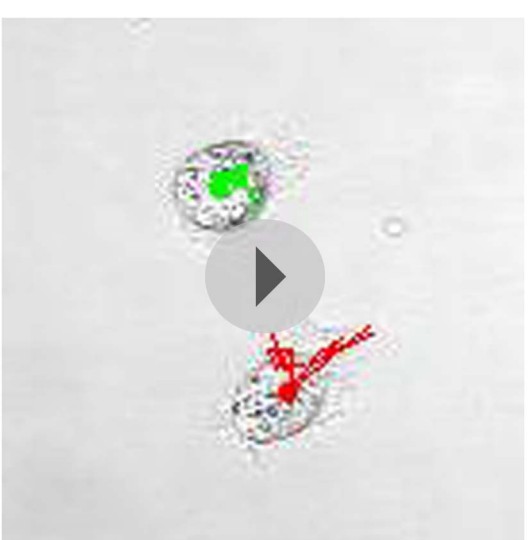

**Video 4.** Live cell imaging and tracking of siATM MDA-MB-231 cells for *Figure 3*. Images were taken every 15 min for 6 hr and tracking was performed in ImageJ. Still images and quantifications are provided in *Figure 3D,E*.

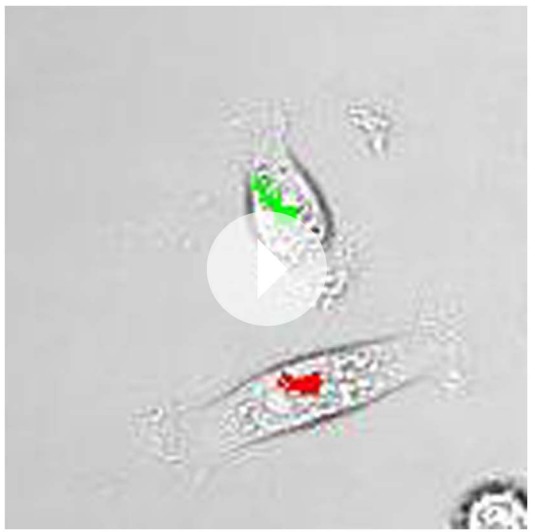

**Video 5.** Live cell imaging and tracking of si mutant p53 MDA-MB-231 cells for *Figure 3*. Images were taken every 15 min for 6 hr and tracking was performed in ImageJ. Still images and quantifications are provided in *Figure 3D,E*.

under normal growth conditions. Nuclear localization of the p50/p65 NF-κB dimer enables transcriptional activation of this complex so we analyzed p65 nuclear accumulation as a readout of NF-κB localization (*Hayden and Ghosh, 2012*). We observed reduced p65 nuclear localization and NEMO phosphorylation in ATM- and mutant p53-depleted cells compared to control cells, which is inline with the reduced *IL-8* expression that occurs under these conditions (*Figure 4G*, *Figure 4—figure supplement 2F*). We next performed chromatin immunoprecipitation (ChIP) of NF-κB on the *IL-8* promoter to analyze directly the involvement of NF-κB in regulating *IL-8* transcription and how this is affected by ATM and mutant p53. ChIP analyses revealed that reduced levels of ATM or mutant p53 impaired NF-κB accumulation on the IL-8 promoter (*Figure 4H*).

Collectively, our results strongly suggest that ATM and mutant p53 are required for NF-κB activity, which is necessary to regulate *IL-8* expression. Further analyses supported the notion of *IL-8* as the gene responsible for reduced migration in ATM-depleted MDA-MB-231 cells as (1) IL-8 depletion reduced cell migration and invasion, (2) NAC treatment reduced *IL-8* mRNA levels and (3) oxidative stress induction by $H_2O_2$ increased *IL-8* levels and (4) $H_2O_2$-induced *IL-8* expression was dependent on ATM (*Figure 5A–E*). Taken together, these results suggest that ATM regulates a transcriptional network that includes the NF-κB-regulated gene *IL-8*. Our data suggests that this ATM pathway promotes cell migration and invasion in MDA-MB-231 cells through a cell intrinsic mechanism that is reliant on endogenous oxidative stress.

## ATM promotes tumor progression in vivo

The importance of IL-8 in promoting cell migration in MDA-MB-231 cells, and its reduction upon ATM inhibition, prompted us to test whether reduced *IL-8* expression in ATM-depleted cells was responsible for reduced migration and invasion in these cells. Supporting this hypothesis, the addition of recombinant IL-8 rescued both migration and invasion properties in both mutant p53 and ATM-depleted cells (*Figure 5F,G*). These results were confirmed with two individual siRNAs targeting ATM to ensure that these results were not due to any siRNA off-target effects (*Figure 5—figure supplement 1*). These data identify *IL-8* as an ATM regulated gene target that strongly influences the reduced migration and invasion of ATM and mutant p53 deficient MDA-MB-231 breast cancer cells.

ATM is considered a tumor suppressor as its deletion in mice results in tumors, patients with mutations in *ATM* in the human disorder Ataxia telangiectasia have increased cancer risks, and several hematological cancers have been reported to express defective ATM (*Cremona and Behrens, 2013*; *Shiloh and Ziv, 2013*; *Stracker et al., 2013*). However, our identification for a requirement of ATM to promote IL-8 expression, cell migration and invasion in vitro, as well as the association of increased levels of IL-8 with the metastatic potential of several cancer types, including colon cancer and melanoma (*Lee et al., 2012*; *Wu et al., 2012*), prompted us to evaluate if ATM could influence tumorigenesis and promote pro-metastatic processes. To address this, we generated a MDA-MB-231 cell line with stable ATM depletion via shRNA. shATM and shNC cells exhibited similar cellular morphology and growth rates in vitro (*Figure 6—figure supplement 1A,B*) and shATM cells exhibited reduced cell migration and invasion, as well as reduced *IL8* expression (*Figure 6—figure supplement 1A,C*). In addition, both shATM and shNC cells formed colonies with similar efficiency in soft agar (*Figure 6A,B*). Thus, ATM depletion by shRNA did not overtly alter the growth properties of MDA-MB0231 cells.

To examine the influence of ATM on tumor growth in vivo, we then orthotopically injected these cells into the mammary fat pad. There, we observed an increase in primary tumor growth, consistent with the well-established role of ATM in tumor suppression (*Figure 6C*). To address whether ATM

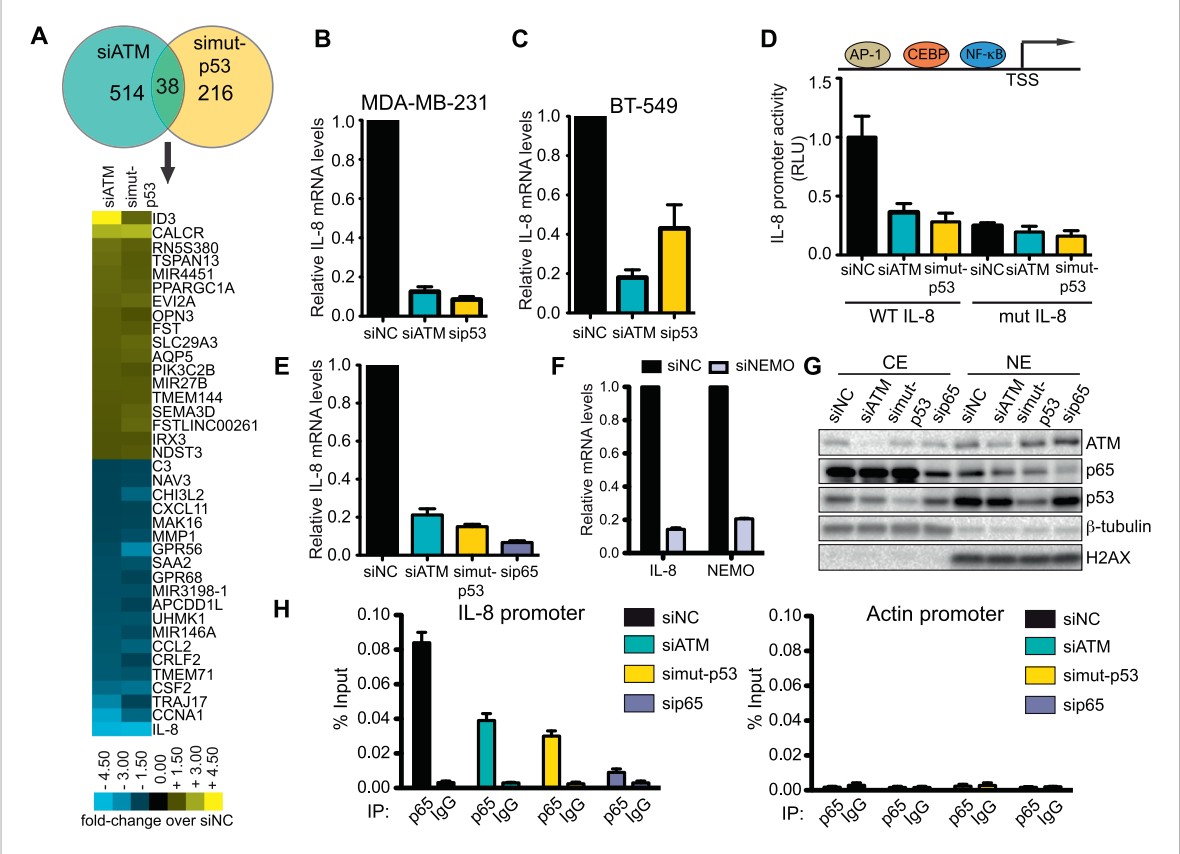

**Figure 4**. ATM-mutant p53 regulates cytokine interleukin-8. (**A**) Differential transcriptome expression analysis in siATM- and simut-p53- depleted cells identifies reduced IL-8 expression in both samples. Upper: Venn diagram of differentially expressed genes in simut-p53, siATM or both. Numbers indicate genes differentially expressed 1.5-fold or greater compared to siNC. Heatmap represents the 38 genes co-regulated similarly in siATM and simut-p53 cells. Expression data was normalized to control siNC cells. Cut-off = 1.5-fold normalized to siNC control cells. (**B** and **C**) qRT-PCR analysis of IL-8 mRNA levels in MDA-MB-231 (**B**) and BT-549 (**C**) siATM or sipmut-53 depleted cells. (**D**) IL-8 promoter activity by luciferase reporter assay in siATM and simut-p53 cells. Depletion of (**E**) NF-κB or (**F**) NEMO impairs IL-8 expression. (**G**) ATM or mutant p53 depletion abrogates NF-κB p65 nuclear localization. Cells treated with indicated siRNA were harvested to obtain cytoplasmic extract (CE) and nuclear extract (NE) to analyze NF-κB p65 localization. (**H**) ATM or mutant p53 deletion impairs NF-κB p65 binding to IL-8 promoter using chromatin immunoprecipitation (ChIP) analysis. Actin promoter serves as a negative control.

The following figure supplements are available for figure 4:

**Figure supplement 1**. Gene ontology (GO) analysis of ATM and mutant p53 regulated genes in MDA-MB-231 cells.

**Figure supplement 2**. Regulation of IL-8 and NF-kB by ATM and p53.

depletion could influence the ability of these cells to colonize secondary sites, we performed tail vein injections with shATM and shNC MDA-MB 231 cells and assessed lung colonization after 9 weeks. In contrast to the pro-tumorigenic effects of ATM depletion on primary growth, lung colonization was strongly suppressed in cells with reduced ATM levels compared to control cells (*Figure 6D,E*), consistent with the ability of ATM activity to regulate IL-8 mediated cell migration and invasion capacity. To examine the relationship between ATM and IL-8 expression and metastasis in human breast cancer, we analyzed a set of expression arrays from 560 breast cancer patients with clinical annotation (*Morales et al., 2014*). The expression of IL-8 was significantly associated with metastasis to the lung but not bone or brain (*Figure 6F*, *Figure 6—figure supplement 2* and *Supplementary file 2*). These data suggest that IL-8, and potentially ATM, do not affect all metastatic processes similarly in all tissues. The mRNA expression levels of ATM were not similarly associated with the probability of metastasis,

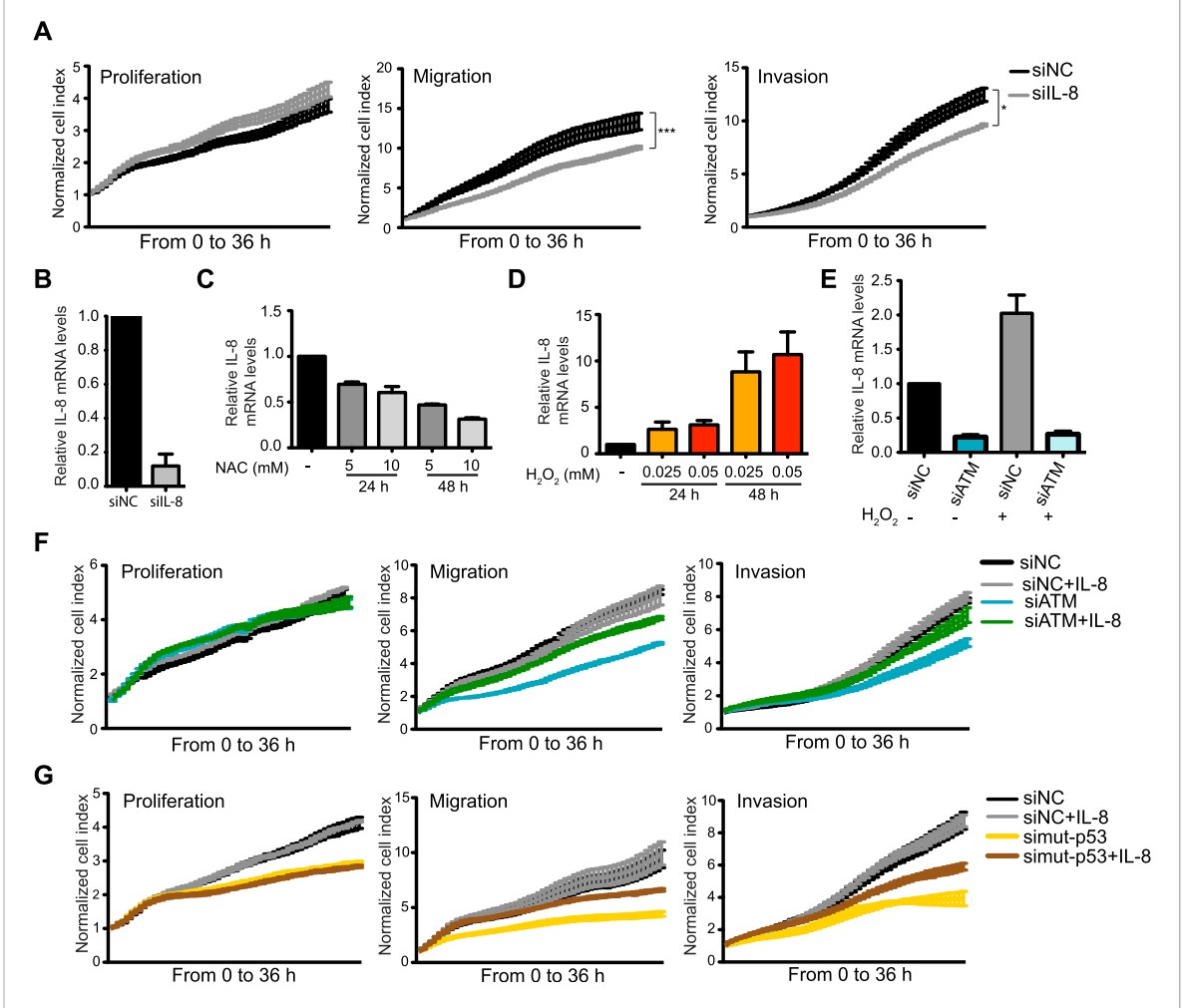

**Figure 5**. ATM promotes pro-metastatic IL-8-dependent cellular processes. (**A**) IL-8 depletion reduces cell migration and invasion. Experiments performed as in *Figure 1G*. Error bars = SEM. * p-value <0.05, *** p-value <0.001, unpaired two-tailed t-test. (**B**) IL-8 qRT-PCR analysis from samples in (**A**). (**C**) ROS inhibitor NAC reduces IL-8 expression. (**D**) $H_2O_2$-induced oxidative stress increases IL-8 expression. (**E**) $H_2O_2$-induced IL-8 expression is dependent on ATM. Cells treated with indicated siRNAs were incubated with 0.025 mM $H_2O_2$ or mock treated and analyzed by qPCR. (**F** and **G**) IL-8 addition restores impaired migration and invasion for ATM-depleted (**F**) or mutant p53-depleted (**G**) cells. Experiments performed as in *Figure 1G* with or without recombinant IL-8. For all graphs, mean ± s.e.m., n = 3.

The following figure supplement is available for figure 5:

**Figure supplement 1**. Exogenous addition of IL-8 rescues migration and invasion defects in ATM-depleted cells.

however, ATM is regulated primarily at the post-transcriptional level and these analyses do not reflect its activity (*Figure 6F*). Collectively, these results demonstrate that ATM can play dual functions in breast cancer tumor progression. On the one hand, ATM serves as a tumor suppressor to limit the proliferation or survival of cancer cells during primary tumor formation. However, based on our findings, ATM can also promote pro-metastatic processes such as lung colonization. These functions of ATM are likely through its ability to control the transcription of key regulators, including IL-8, that promote cell migration and invasion to influence metastatic colonization.

## Discussion

In summary, we have identified a tumor promoting activity for ATM. Our data is consistent with this pathways being triggered by intrinsic oxidative stress that occurs in highly metastatic cancer cells. This

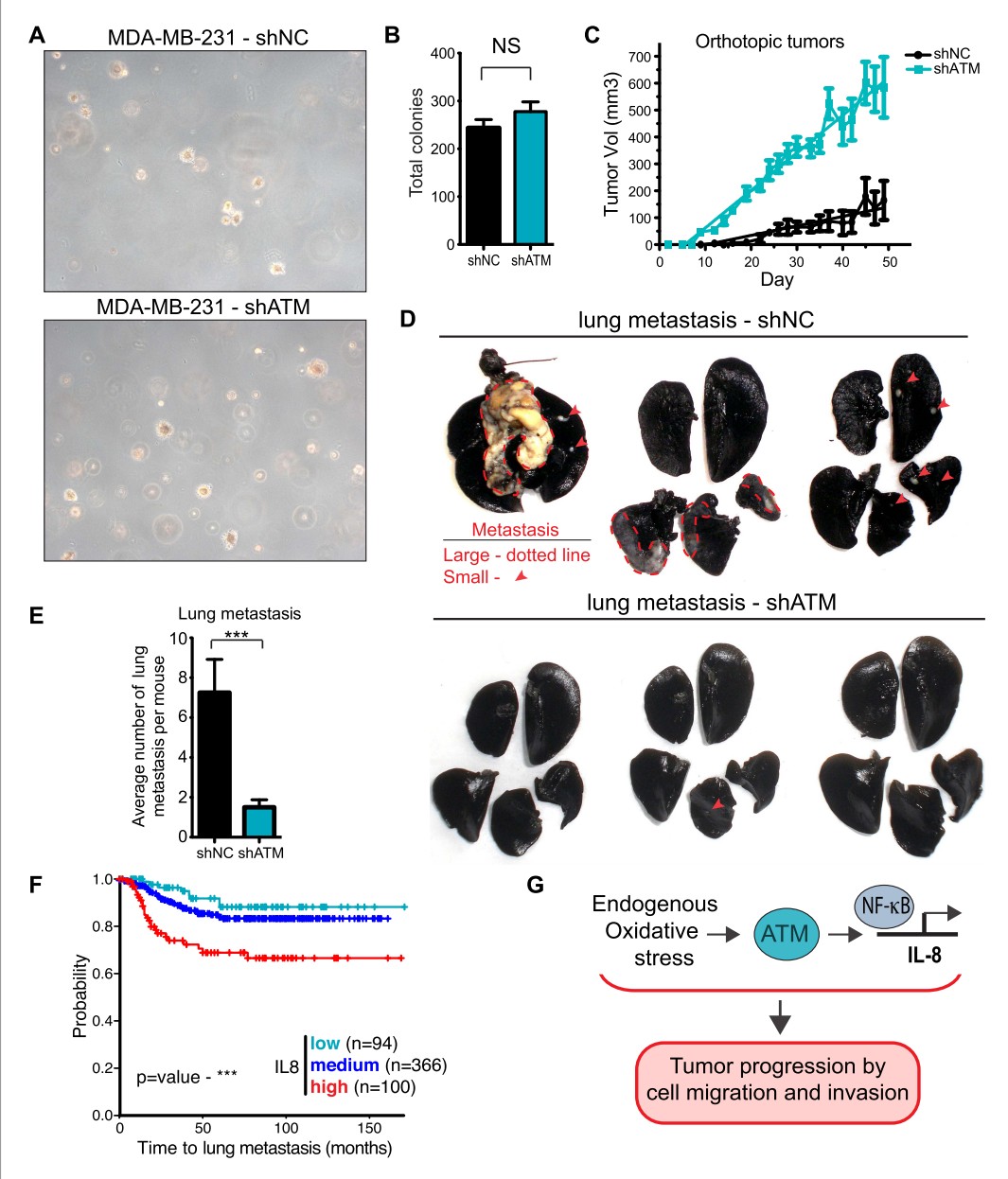

**Figure 6**. ATM promotes tumor progression in vivo. (**A**) ATM is not required for colony growth in soft agar. Representative images of shNC and shATM are shown. (**B**) Quantification of soft agar assays. Colonies were counted from 10 fields of view. Differences between shNC and shATM are not significantly different (NS) by unpaired two-tailed t-test. (**C**) MDA-MB-231 shATM cells increase cancer cell proliferation compared to shNC cells by measuring orthotopic tumor growth. (**D**) ATM-depletion reduces lung tumor formation. Representative India ink stained lungs at necropsy from mice (N = 12 for each group) following tail-vein injection xenografting of shNC or shATM MDA-MB-231 cells. Lung tumors indicated by arrows and dotted lines. (**E**) Quantification of lung tumors from **D**. (**F**) Kaplan-Meier plot of probability of lung metastasis free survival in 560 breast cancer patients based on IL8 expression levels. (**G**) Model for ATM pathway in tumor progression.

The following figure supplements are available for figure 6:

**Figure supplement 1**. ATM is required for cell motility and IL-8 expression.

**Figure supplement 2**. Analysis of gene expression data from breast cancer patients.

ATM-dependent pathway regulates the transcription of pro-metastatic genes including *IL-8* through the activation of the transcriptional regulator NF-κB. The expression of these genes, including *IL-8*, results in increased cell migration and invasion that aid in tumor progression (*Figure 6G*). ATM is known to regulate several pro-inflammatory cytokines as part of a senescence-associated secretory phenotype (SASP) in senescent cells (*Rodier et al., 2009*; *Coppe et al., 2010*). The SASP can be mediated by DNA damage induced ATM signaling through MRN and CHK2, but not p53. In addition, a role for p38 MAPK in the SASP has been defined that is independent of the initial ATM-dependent damage response (*Freund et al., 2011*). Senescent cells are thought to promote tumor progression in the microenvironment through paracrine signaling by secreted pro-inflammatory factors including cytokines (*Coppe et al., 2008*, *2010*). ATM has also been shown to regulate IL-8 secretion in response to DSB formation by etoposide, which is dependent on NF-κB and NEMO but independent from p53 (*Biton and Ashkenazi., 2011*). From our data, we conclude that ATM, through a pathway requiring mutant p53, can support a transcriptional program that includes several pro-metastatic genes including IL-8 that maintain cell migration and invasion. These processes are operational in cancer cells and have the ability to support cellular activities contributing to tumorigenesis. Interestingly, a recent study identified a fundamentally different type of anti-cancer pathway mediated by ATM in a MLL-AF9 driven acute myeloid leukemia model (*Santos et al., 2014*). In this instance, ATM was required to block the differentiation capacity of these cells, thereby promoting their tumorigenic properties. p38 MAPK plays both pro and anti-tumorigenic roles and its downregulation has been demonstrated to reduce tumor growth and survival but also to promote the metastasis of colorectal cancer to the lung through the modulation of inflammatory responses (*Gupta et al., 2014*; *Urosevic et al., 2014*). Thus, emerging evidence indicates that ATM represents another DDR factor that, like p53 or p38, can differentially influence pathways relevant to diverse aspects of tumorigenesis in a context dependent manner.

Our study emphasizes the ability of ATM to regulate the transcription of a sub-set of genes involved in tumor promoting pathways. Comparative expression analysis revealed wound healing as a major pathway regulated by both ATM and mutant p53. Several pro-metastatic genes from this analysis in addition to *IL-8*, including *CXCL11*, *CCL2*, *MMP1* and *SAA2*, were identified. By identifying *IL-8* as a target of oxidative stress and ATM, we provide a link between activated ATM by oxidative stress and pro-metastatic pathways operating in cancer cells. Our results provide additional insights into other disease pathways involving ATM including angiogenesis, a pathway triggered by oxidative stress and dependent on IL-8 (*Koch et al., 1992*; *Okuno et al., 2012*). Inhibitors of the IL-8 pathway are actively being developed and evaluated in pre-clinical studies and clinical trials for cancer and inflammatory diseases (*Waugh and Wilson, 2008*; *Singh et al., 2013*). Our work provides a rationale for evaluating the clinical efficacy of IL-8 inhibition in the context of lung metastasis, including the use of ATM inhibitors beyond their ability to radio-sensitize cancer cells. It is intriguing that high IL-8 expression levels in breast cancers were specifically correlated with lung metastasis, a tissue location with high oxygen burden (*Rosanna and Salvatore, 2012*). Thus, regulation of IL-8 by ATM could promote a metastatic program that is important in an organ-specific context where oxidative stress and cytokine secretion occur. We hypothesize based on our findings that targeting ATM could inhibit IL-8 dependent processes involving tumor progression and metastasis in certain cancers. In addition, based on our study and those of others, ATM inhibition could also be efficacious in the setting of other diseases involving oxidative stress and pro-inflammatory cytokines including inflammation, angiogenesis and aging (*Osorio et al., 2012*; *Okuno et al., 2012*).

## Materials and methods

### Cell culture, reagents and treatments

MDA-MB-231, MCF-7, and U2OS cells were grown in Dulbecco's Modified Eagle's Medium (DMEM) supplemented with 10% fetal bovine serum (FBS), 100 U/ml penicillin, 100 mg/ml streptomycin and 2 mM L-glutamine and BT-549 cells were maintained in RPMI-1640 medium with the same supplements. DNA-damaging treatments used in wound-healing assay were as follows: ATM inhibitor KU-55933 (ATMi, 10 μM), doxorubicin (Dox, 100 nM), phleomycin (Phleo, 90 μg/ml), etoposide (Et, 40 μM), and IR (20 Gy) by a Faxitron X-ray unit, (Faxitron X-ray Corporation, Tucson, AZ). Cells were incubated with these agents for 48 hr except IR, which is imposed on cells at 0 hr followed by incubating the cells with regular medium.

## Western blotting analysis, ChIP and subcellular fractionation

Cells were washed once with PBS (phosphate buffered saline), and lysed in Laemmli buffer (4% SDS, 20% glycerol and 120 mM Tris, pH 6.8) to obtain whole cellular lysate. The cell lysates were sonicated in a Diagenode Biorupter 300 for 10 min, boiled for 5 min at 95°C followed by centrifugation before loading. Samples were separated by SDS-PAGE and detected using standard chemiluminescence (GE Healthcare Amersham ECL prime, Piscataway, NJ) using a Bio-Rad Molecular Imager ChemiDoc XRS+ system. For ChIP, cells were crosslinked in 1% formaldehyde for 10 min followed by 125 mM glycine treatment to stop the reaction. Cells were then lysed in RIPA buffer (50 mM Tris-HCl, pH 8.0, 150 mM NaCl, 2 mM EDTA, pH 8.0, 1% NP-40, 0.5% Sodium Deoxycholate, 0.1% SDS and a proteinase inhibitor) and sonicated in a Biorupter (Diagenode, Denville, NJ). Samples were immunoprecipitatd with NF-κB p65 antibody overnight at 4°C. After incubation of protein A agarose for 4 hr at 4°C, beads were washed once with the following solutions: TSE-150 (1% Triton X-100, 0.1% SDS, 2 mM EDTA, 20 mM Tris-HCl, pH 8.0, 150 mM NaCl), TSE-500 (1% Triton X-100, 0.1% SDS, 2 mM EDTA, 20 mM Tris-HCl, pH 8.0, 500 mM NaCl), LiCl buffer (0.25 M LiCl, 1% NP-40, 1% SOC, 1 mM EDTA, 10 mM Tris, pH 8.0), and TE buffer (10 mM Tris pH 8.0, 0.1 mM EDTA pH 8.0). After reverse crosslinking, DNA was purified using a PCR purification kit (Qiagen, Netherlands and Germany). qRT-PCR analysis was performed using SYBR green (Applied Biosystems, Foster City, CA) on an Applied Biosystems StepOnePlus system. For cytoplasmic and nuclear extract, $3 \times 10^6$ cells were suspended in 150 µl solution containing 10 mM HEPES, pH7.9, 10 mM KCl, 1.5 mM $MgCl_2$, 0.34 M sucrose, 10% glycerol, 1 mM DTT, 0.1% TritonX-100 and a proteinase inhibitor. Followed by incubating on ice for 5 min, cells were harvested by centrifugation at 1300×$g$ for 4 min to acquire cytoplasmic (supernatant) and nuclear (pellet) fractions. Then the pellet was added 200 µl of Laemmli buffer and sonicated for 10 min. These extracts were analyzed as above described. Antibodies used for western blotting were as follows: H2AX (#2595; Cell Signaling, Beverly, MA), γH2AX (#NB100-384; Novus Biologicals, Littleton, CO), Phospho-p53 (Ser15) (#9284; Cell Signaling), p53 (#39553; Active Motif, Carlsbad, CA), ATM (#sc-135663; Santa Cruz Biotechnology, Santa Cruz, CA), CHK2 (#2662; Cell Signaling), KAP1 (#sc-33186; Santa Cruz Biotechnology), phosphor-KAP1 Ser824 (#A100-767A; Bethyl Laboratories, Montgomery, TX), MRE11 (#ab214; Abcam, United Kingdom), NBS1 (#NB100-143; Novus Biologicals), NF-κB p65 (#sc-109; Santa Cruz Biotechnology), NF-κB p65 Ser536 (#3033; Cell Signaling), NF-κB p65 Ser468 (#3039; Cell Signaling), NF-κB p65 Ser276 (#sc-101749; Santa Cruz Biotechnology), NEMO (#sc-8330; Santa Cruz Biotechnology), NEMO Ser85 (#ab63551; Abcam), and β-tubulin (#ab6046; Abcam). Secondary antibodies conjugated to horseradish peroxidase (Cell Signaling) were used for enhanced chemiluminescence.

## siRNA transfection

Small interfering RNA (siRNA) SMARTpools were obtained from Dharmacon: siNC (Non-targeting pool), siATM, siCHK2, siMRE11, siNBS1 and sip53. The sequences for independent siATM-1 and siATM-2, which are part of siATM SMARTpools, are GCAAAGCCCUAGUAACAUA and GGGCAUUACGGGUGUUGAA (Dharmacon, Lafayette, CO). The sequence for siIL-8 is GCCAAGGA-GUGCUAAAGAA (Dharmacon). The sequence for sip65 is GAUCAAUGGCUACACAGGAUU (*Dai et al., 2005*). The sequence for siNEMO is GGAAGAGCCAACUGUGUGAUU (*Hinz et al., 2010*). Lipofectamine RNAiMAX (Invitrogen, Carlsbad, CA) was used to transfect the indicated siRNA into cells following the manufacturer's instructions.

## Luciferase reporter assay

The promoter region of IL-8 corresponding to −1423/+96 bp was cloned from MDA-MB-231 cells by PCR using 5′-CTCGAGGTAACCCAGGCATTATTTTA-3′ and 5′-AGATCTAGCTTGTGTGCTCTGCTGTC-3′ primers. The PCR fragment was then subcloned into pCR-Blunt II-TOPO vector (Invitrogen). The IL-8 promoter was cut with XhoI and BglII and cloned into Xho1/BglII treated pNANOG-Luc (Addgene No. 25900). After sequencing, this IL-8 promoter driven luciferase construct was named as pIL8-Luc. As a negative control, we made another IL-8 reporter with mutated NF-κB binding (named as mut pIL8-Luc) by using pIL8-Luc as a template and the following primers:

> 5′-ATGGGCCATCAGTTGCAAATCGTTAACTTTCCTCTGACATAATGAAAAGA-3′
> 5′-TCTTTTCATTATGTCAGAGGAAAGTTAACGATTTGCAACTGATGGCCCAT-3′

The luciferase assay was performed by transfecting with indicated siRNA on day 1. After 24 hr, pIL8-Luc or mut pIL8-Luc and a trace amount of pRL-SV40P as an internal control (Addgene No. 27163)

were transfected into cells with FugeneHD (Promega, Madison, WI). On day 3 or 4, cells were lysed and both firefly and renilla luciferase were measured by using Dual-Glo Luciferase assay system (Promega) on a microplate reader (TECAN, Switzerland). All firefly luciferase measurements were corrected for renilla luciferase values. The ratios were normalized to control groups and expressed as relative luciferase unit.

## Wound-healing, soft agar and live-cell imaging assays

MDA-MB-231 and BT-549 cells were seeded into 6-well plates and cultured to confluence. Cells were then wounded with sterile pipette tips and washed with PBS. Then cells were treated or untreated with DNA-damaging agents as described. For wound-healing assay with siRNA treatments, cells were transfected with indicated siRNA and after 48 hr, wounds were made as described above. Pictures were acquired at 0 hr and 48 hr using an EVOS fl fluorescence microscope. Representative pictures were from three independent experiments and quantified by using TScratch software (*Geback et al., 2009*). Soft agar assays were performed on shNC and shATM MDA-MB-231 cells essentially as described (*Xhemalce et al., 2012*). For live analysis of cell motility, MDA-MB-231 cells were plated in glass round petri dish (TED PELLA, INC, Redding, CA) and treated as indicated. Images were taken every 15 min using Olympus FluoView FV1000 confocal microscope and analyzed with ImageJ manual tracking plugin. Greater than 100 cells were analyzed for each experiment and sample and results provided are from three independent experiments.

## Cell proliferation assay

Cellular proliferation of MDA-MB-231 was counted by trypan blue staining. Briefly, cells were plated in 6-well plates and treated with indicated DNA-damaging agents. Then, the cells were trypsinized and counted by trypan blue staining.

## Flow cytometry

For cell-cycle analysis, MDA-MB-231 cells untreated or treated with indicated DNA-damaging agents for 48 hr, except that cells were exposed to IR at 0 hr followed by incubated with regular medium, were trypsinized and fixed in 70% ice-cold ethanol overnight at 4°C. Cells were centrifuged, suspended and incubated in PBS containing propidium iodide (50 µg/ml) and RNase A (100 µg/ml) for 30 min at room temperature. DNA content was analyzed by using BD Accuri C6 flow cytometer, and FlowJo software was used to analyze flow cytometry data. Intracellular ROS was determined by CM-H$_2$DCFDA (Invitrogen). Briefly, following indicated treatments, cells were trypsinized and incubated with 5 µM CM-H$_2$DCFDA in PBS for 30 min in the 37°C incubator. Cells were then returned to phenol red-free medium for 15 min recovery period and immediately analyzed by BD Accuri C6 flow cytometer. 4 mM H$_2$O$_2$ treatment served as a positive control and 10 mM NAC was used as a reducing agent. Data was analyzed using FlowJo and mean fluorescence intensity was used as a measure of ROS.

## RTCA real-time analysis of proliferation, migration and invasion

xCELLigence Real-Time Cell Analyzer (RTCA; Roche Diagnostics, Switzerland) was used to monitor cell proliferation, migration and invasion independently in a label-free, real-time setting. When cells contact and adhere to electrical sensors, this leads to increasing electrical impedance. The impedance correlates with an increase in proliferating, migrated or invaded cells, derived as a parameter called cell index. For proliferation experiment, cells were treated as indicated and seeded in quadruplicates on E-plate. For migration and invasion assays, following indicated treatments, cells were starved in serum-free medium overnight and seeded in quadruplicates on CIM-plate. Matrigel (BD Biosciences) was diluted in serum-free medium at a ratio of 1:40 and coated on CIM-plate only for invasion. Regular growth medium was added in the lower chamber as chemoattractant. The cell index was measured every 15 min, and the results were represented as normalized cell index. For exogenous addition of interleukin-8 (IL-8) (PEPROTECH), 150 ng/ml of IL-8 was added to cells transfected with siATM and 200 ng/ml of IL-8 was added to cells depleted for p53.

## Microarray analysis

Total RNA form cells transfected with siNC, siATM or sip53 was extracted using RNeasy Mini kit (Qiagen) following manufacturer's instructions. All RNA was DNase treated before sendind to Microarray core facility at DANA-Farber Cancer Institute to perform Affymetrix GeneChip Human Gene 2.0 ST array (n = 2 in each group). Affymetrix Expression Console (EC) was used to generate CHP files from CEL files. Then CHP files were loaded into Affymetrix Transcriptome Analysis Concole (TAC) software. Normalizaton and gene expression analysis were performed in TAC software. GO analysis was analyzed using the 'Functional Annotation Tool' in DAVID (http://david.abcc.ncifcrf.gov/home.jsp) and biological process terms are shown (*Huang da et al., 2009*).

## Quantitative real-time PCR

Total RNA from cells treated as indicated was purified using RNeasy Mini kit (Qiagen) and treated with DNase according to manufacturer's instructions. 1 µg of total RNA was used for cDNA systhesis with SuperScript III first-strand synthesis system. To analyze IL-8 mRNA expression, we designed gene-specific qPCR primers: 5′-AAGAAACCACCGGAAGGAAC-3′ and 5′-ACTCCTTGGCAAAACTGCAC-3′ for IL8. GAPDH (glyceraldehyde-3-phosphate dehydrogenase) (Quantitect primer assay, Qiagen) was used for normalization. For analyzing NEMO and p65 expression, we designed the following qPCR primers: 5′-AGAGTCTCCTCTGGGGAAGC-3′ and 5′-GCTTGGAAATGCAGAAGCTC-3′ for NEMO; 5′-ACAACCCCTTCCAAGTTCCT-3′ and 5′-ATCTTGAGCTCGGCAGTGTT-3′ for p65. To perform ChIP-qPCR, we used the following primers: (1) 5′-GTGTGATGACTCAGGTTTGC-3′ and 5′-GTTTGTGCCTTATGGAGTGC for IL-8 promoter and (2) 5′-CGGAAAGATCGCCATATATGGAC-3′ and 5′-ACCGGCAGAGAAACGCGA-3′ for actin promoter. Quantitative real-time PCR analysis was performed using SYBR green (Applied Biosystems) on an Applied Biosystems StepOne Plus system.

## Mouse lung metastasis and orthotopic tumor growth experiments

Stable MDA-MB-231 shNC and shATM cells were made by using shATM (sequence: GAAGTA-GAAGGAACCAGTTACCATGAATC) and shNC (control) plasmids from Origene. These cells were maintained in regular DMEM with 0.2 µg/ml puromycin. Balb/c mice were maintained according to the University of Texas at Austin Institutional Animal Care and Use Committee guidelines. $0.5 \times 10^6$ shNC or shATM MDA MB 231 breast cancer cells were injected into the mouse tail vein. Mice were then monitored daily for symptoms of distress, including difficulty breathing, ruffled appearance, and limited mobility. Surviving mice were euthanized 9 weeks after cell injections. Mouse lungs were perfused with India ink to quantify metastatic lesions as previously reported (*Williams et al., 2004*). For orthotopic tumor experiment, $2 \times 10^6$ shATM or shNC MDA MB 231 cells were injected into the #4 mammary fat pad of 6 week-old Balb/c mice. Mice were palpated three times weekly until tumors reached 750 mm$^3$. Tumor size was expressed as tumor volume (mm$^3$) and calculated by the formula: volume = (smaller dimension$^2$ × larger dimension)/2.

## Analysis of breast cancer patient gene expression data

Publically available breast cancer patient gene expression data from GEO was pooled (GSE2603, GSE2034, GSE5327, and GSE12276) as previously described (*Minn et al., 2005*; *Barrett et al., 2007*; *Morales et al., 2014*). In order to remove systematic biases, expression measurements were converted to *z*-scores for all genes before merging. Multiple probes from the same gene were converted to a mean, patients were grouped based on expression levels and Kaplan–Meier survival was plotted. The hazard ratio (HR) and p value for each gene was calculated using a Cox proportional hazards model and performing likelihood ratio tests. The HR was checked for consistency over time, fulfilling assumptions of the Cox model. All significance measurements were done using expression as a continuous variable.

## Acknowledgements

We thank the Miller lab for helpful discussions. We are grateful to T Paull for critical reading of the manuscript and to Camille Stephan-Otto Attolini and the IRB Barcelona Biostatistics/Bioinformatics core facility. THS is supported by a grant from the Ministerio de Economía y Competitividad (BFU2012-39521). The research in the KMM laboratory was supported in part by start-up funds from

the University of Texas at Austin and the Cancer Prevention Research Institute of Texas (CPRIT, R116). KMM is a CPRIT scholar.

## Additional information

### Funding

| Funder | Grant reference | Author |
|---|---|---|
| Ministerio de Economía y Competitividad | BFU2012-39521 | Travis H Stracker |
| Cancer Prevention and Research Institute of Texas (CPRIT) | R116 | Kyle M Miller |

The funders had no role in study design, data collection and interpretation, or the decision to submit the work for publication.

### Author contributions

W-TC, THS, CLVDB, Conception and design, Acquisition of data, Analysis and interpretation of data, Drafting or revising the article; NDE, Acquisition of data, Analysis and interpretation of data; BX, Conception and design, Acquisition of data, Analysis and interpretation of data; KMM, Conception and design, Analysis and interpretation of data, Drafting or revising the article

### Ethics

Animal experimentation: Experiments involving Balb/c mice for this study were performed in strict accordance with guidelines set forth for the handling and care of animals by the institutional animal care and use committee (IACUC) protocols (AUP-2012-00075) of the University of Texas at Austin.

## Additional files

### Supplementary files

• Supplementary file 1. Gene expression analysis. (**A**) Differentially expressed genes in siATM MDA-MB-231 cells. MDA-MB-231 cells depleted for ATM were analyzed for gene expression using Affymetrix GeneChip Human Gene 2.0 ST array. The gene list was generated from Affymetrix Transcriptome Analysis Concole (TAC) software. (**B**) Differentially expressed genes in si mutant p53 MDA-MB-231 cells. This gene list is generated as in A. (**C**) Genes regulated by both ATM and mutant p53 in MDA-MB-231 cells. This gene list was generated from data in A and B by choosing genes regulated similarly by both ATM and mutant p53.

• Supplementary file 2. Statistical analysis of IL8 expression in metastasis to the indicated tissue. Univariate Cox proportional hazard regression for continuous variables (first row of table) and log-rank test of equality across strata for categorical variables (grouping of samples according to IL8 expression).

### Major datasets

The following previously published datasets were used:

| Author(s) | Year | Dataset title | Dataset ID and/or URL | Database, license, and accessibility information |
|---|---|---|---|---|
| Minn AJ, Massague J | 2005 | Subpopulations of MDA-MB-231 and Primary Breast Cancers | http://www.ncbi.nlm.nih.gov/geo/query/acc.cgi?acc=GSE2603 | Publicly available at NCBI Gene Expression Omnibus (Accession no: GSE2603). |
| N/A | 2005 | Breast cancer relapse free survival | http://www.ncbi.nlm.nih.gov/geo/query/acc.cgi?acc=GSE2034 | Publicly available at NCBI Gene Expression Omnibus (Accession no: GSE2034). |

| Author(s) | Year | Dataset title | Dataset ID and/or URL | Database, license, and accessibility information |
|---|---|---|---|---|
| Wang Y, Foekens J, Minn A, Massague J | 2007 | Breast cancer relapse free survival and lung metastasis free survival | http://www.ncbi.nlm.nih.gov/geo/query/acc.cgi?acc=GSE5327 | Publicly available at NCBI Gene Expression Omnibus (Accession no: GSE5327). |
| N/A | 2009 | Expression data from primary breast tumors | http://www.ncbi.nlm.nih.gov/geo/query/acc.cgi?acc=GSE12276 | Publicly available at NCBI Gene Expression Omnibus (Accession no: GSE12276). |

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
