## [Decision Letter]

Thank you for sending your work entitled “ATM regulation of IL-8 links oxidative stress to cancer cell migration and invasion” for consideration at *eLife*. Your article has been favorably evaluated by James Manley (Senior editor) and two reviewers, one of whom is a member of our Board of Reviewing Editors.

The Reviewing editor and the other reviewer discussed their comments before we reached this decision, and the Reviewing editor has assembled the following comments to help you prepare a revised submission:

In their manuscript, Chen et al. describe an interesting dataset revealing an oncogenic, pro-metastatic role for the protein kinase ATM. Using breast cancer cells, they report that ATM is required for cell migration in a ‘wound healing’ experimental system, that this function of ATM is independent of its canonical role in the DNA damage response (DDR), acting instead through an alternative ROS-mediated activation pathway. They report that mutant p53 also contributes to cell migration in this system. Gene expression analysis downstream of ATM and mutant p53 reveals that both factors are positive regulators of IL-8 expression in a mechanism likely involving the transcription factor NF-κB. In turn, they demonstrate that IL-8 partially mediates the pro-invasive roles of ATM and mutant p53 in their system. They also show data from mouse xenograft experiments demonstrating a pro-metastatic role for ATM.

Overall, the manuscript displays great scholarship, with a strong dataset, and conclusions are fully supported by the data. Over the past few years it has become apparent that ATM plays other roles outside of its canonical role as the apical kinase in the DDR, and that some of these non-canonical roles may be oncogenic rather than tumor suppressive. Several lines of evidence point to a role for ATM as a sensor of oxidative stress playing roles in mitophagy, ROS control and central metabolism. Within this context, the pathway described here (i.e. ROS>ATM>NF-κB>IL8>invasion/metastasis) is deemed significant. The data are convincing and the connection to lung metastasis in particular is intriguing, especially given the high oxygen burden in lung in particular.

Overall, the reviewers assessed the manuscript as potentially meritorious of publication in *eLife* after addressing the following major concerns:

The reviewers pointed out that some aspects of the molecular pathway elucidated have been previously reported including the role of ATM in regulation of IL-8 production via NF-κB (Costa et al. Plos Pathog. 2013, PMID:24068928) and the ability of ATM to promote migration in a ROS-dependent manner (Singh et al. Cell Signal. 2014, PMID: 24705025). Additionally, pro-migratory roles for various p53 mutants have been well established (Muller and Vousden, Cancer Cell 2014, PMID: 24651012). In order to increase the impact and novelty of this work, the reviewers recommend additional experiments that would raise mechanistic insight and significance.

1) The authors demonstrate that ATM, mutant p53, and p65, are all involved in the regulation of IL-8 and that p65 associates with the IL-8 promoter. What is lacking is an explanation of exactly how ATM and mutant p53 are involved in this process. Does ATM directly phosphorylate mutant p53 or p65? Is this effect via the established NEMO-RIPK1 complex? Does mutant p53 bind to the IL-8 promoter? Is mutant p53 associated with p65 in the cytoplasm and preventing its nuclear import? A more thorough investigation of the mechanism of IL-8 regulation would improve this manuscript significantly.

2) ATM is known to regulate NF-κB signals at multiple levels, especially through direct phosphorylation of NEMO. The epispastic relationship between ATM and p53 in IL-8 regulation is intriguing and could have important implication in lung metastasis. Given both MDA-MB-231 or BT549 cell lines express mutant versions of p53 (R280K for MDA-MB-231 and R249S for BT549), it would be important to determine whether ATM regulates IL-8 expression in the context of WT p53 protein or only in the presence of mutant p53.

3) Given ATM regulates cell migration and IL-8 expression independent of DNA damage, it would be very helpful to determine whether ATM kinase activity is required for cell migration using the specific kinase inhibitors.

4) ATM is known to promote NF-κB signal, an important regulator of IL-8 expression. In this context, it is helpful to determine the impact of mutant p53 on NF-κB signal (e.g. phosphorylation status NEMO or levels and phosphorylation of p65), whether both ATM and mutant p53 promotes NF-κB signal to regulate IL-8 expression.

---

## [Author Response]

*The reviewers pointed out that some aspects of the molecular pathway elucidated have been previously reported including the role of ATM in regulation of IL-8 production via NF-κB (Costa et al. Plos Pathog. 2013, PMID:24068928) and the ability of ATM to promote migration in a ROS-dependent manner (Singh et al. Cell Signal. 2014, PMID: 24705025)*.

We agree that some aspects of our work are related to those reported in these publications. In Costa et al., data was presented that induction of DNA damage by a viral protein induced the secretion of IL-8, which was ATM-dependent. We believe that our data suggest a very different scenario whereby endogenous oxidative stress results in ATM activation, which induces the transcription of IL-8. In Singh et al., macrophage conditioned medium was shown to induce cell migration in an ATM-dependent manner in MCF7 cells. MCF7 cells contain wild-type p53, and we now provide new data showing that ATM does not promote IL-8 expression levels in a wild-type p53 background, see Figure 4—figure supplement 1. In addition, IL8 was not studied in Singh et al. Thus, the pathway described in Singh et al. is likely to be very different in many aspects from our study. Therefore, although one can draw similarities between these studies, we believe our work is clearly novel and complimentary.

*Additionally, pro-migratory roles for various p53 mutants have been well established (Muller and Vousden, Cancer Cell 2014, PMID: 24651012)*.

We completely agree and also cited this same review in the Introduction of our manuscript. We believe the significance of our work is the identification of ATM in a pro-migratory oxidative stress induced pathway involving IL-8 that promotes oncogenic functions, a finding that is significant and novel. Our data suggests that ATM is involved with these well-established pro-migratory roles of mutant p53, which is an exciting new finding.

*In order to increase the impact and novelty of this work, the reviewers recommend additional experiments that would raise mechanistic insight and significance*.

*1) The authors demonstrate that ATM, mutant p53, and p65, are all involved in the regulation of IL-8 and that p65 associates with the IL-8 promoter. What is lacking is an explanation of exactly how ATM and mutant p53 are involved in this process. Does ATM directly phosphorylate mutant p53 or p65? Is this effect via the established NEMO-RIPK1 complex? Does mutant p53 bind to the IL-8 promoter? Is mutant p53 associated with p65 in the cytoplasm and preventing its nuclear import? A more thorough investigation of the mechanism of IL-8 regulation would improve this manuscript significantly*.

IL-8 is known to be a transcriptional target of NF-κB and we have identified reduced NF-κB binding to the IL-8 promoter in ATM and mutant p53 depleted cells. These results are consistent with our observation that IL-8 levels are highly reduced under these conditions. Upon DNA damage, ATM is known to phosphorylate p53 on S15 which is involved in promoting its activity. NEMO is also phosphorylated by ATM upon DNA damage which activates the NF-κB pathway. How these pathways operate in cancer cells with high ROS levels is unknown. To further investigate how IL-8 is regulated by these pathways we have performed several additional experiments as suggested by the reviewers. We have found that ATM can phosphorylate p53 on S15 in MDA-MB-231 cells Figure 3—figure supplement 1 suggesting that ATM could regulate mutant p53 directly. We have also analyzed the role of NEMO in regulating IL-8. Depletion of NEMO resulted in reduced IL-8 mRNA levels, the same result obtained in siATM or simutantp53 cells. These data demonstrate that NEMO is required for this pathway, data consistent with its established role in activating NF-κB. We have included these new data in Figure 4.

We have also performed several additional experiments to determine how mutant p53 could regulate NF-κB. We tested whether mutant p53 associates with the IL-8 promoter. Unlike NF-κB, we were unable to observe enrichment of mutant p53 on the IL-8 promoter (data not shown). Although not definitive given this negative result, our inability to identify mutant p53 bound to the IL-8 promoter suggests that mutant p53 functions within a different context to regulate the promoter activity of IL-8. We also tested whether mutant p53 could interact with NF-κB in MDA-MB-231 cells. co-IP western experiments with mutant p53 or p65 did not detect any physical interaction between these proteins in MDA-MB-231 cells (data not shown). This suggests that it is perhaps unlikely that mutant p53 interacts with p65 in the cytoplasm to inhibit NF-κB activity. We instead favor a model whereby mutant p53 affects NEMO activation. We have new data showing that depletion of mutant p53 results in reduced NEMO phosphorylation (see 4 below). This suggests that mutant p53 helps promote NEMO activation, which is a known modulator of NF-κB. How gain-of-function p53 mutations alter p53 to promote pro-tumorigenic functions is known to be complex. Our additional experiments have identified the requirement of NEMO in regulating IL-8 and suggest that mutant p53 participates in this pathway. From these lines of evidence, we conclude that mutant p53 does not regulate IL-8 directly by binding to its promoter or through direct interactions with p65. We are excited to follow-up on these results in detailed and extensive future studies to analyze in-depth the molecular mechanisms that govern the crosstalk between ATM, mutant p53 and NF-κB signaling pathways, especially in regards to promoting cellular migration and invasion in response to oxidative stress.

*2) ATM is known to regulate NF-κB signals at multiple levels, especially through direct phosphorylation of NEMO. The epispastic relationship between ATM and p53 in IL-8 regulation is intriguing and could have important implication in lung metastasis. Given both MDA-MB-231 or BT549 cell lines express mutant versions of p53 (R280K for MDA-MB-231 and R249S for BT549), it would be important to determine whether ATM regulates IL-8 expression in the context of WT p53 protein or only in the presence of mutant p53*.

Several previous studies have shown that WT p53 represses transcription of IL-8 (ex. Hidaka et al., J Exp Clin Cancer Res 2005, PMID: 15945132 and Yeudall et al., Carcinogenesis 2012, PMID: 22114072). Similar to WT p53, we observed that IL-8 mRNA levels increased upon siRNA-depletion of ATM in two different human cancer cell lines containing WT p53, U2OS and MCF7, respectively. These data support the idea that ATM functions to promote IL-8 expression in the context of gain-of-function mutant p53. We have included these new data in Figure 4—figure supplement 2 in the revised manuscript.

*3) Given ATM regulates cell migration and IL-8 expression independent of DNA damage, it would be very helpful to determine whether ATM kinase activity is required for cell migration using the specific kinase inhibitors*.

As suggested, we tested whether an ATM kinase inhibitor could affect cell migration. Consistent with our genetic studies of ATM depletion by siRNA or shRNA, treatment of MDA-MB-231 cells with the specific ATM kinase inhibitor KU-55933 resulted in reduced cell migration (Figure 1—figure supplement 1). This data suggests that ATM kinase activity is required for cell migration. We now include this new data along with additional text describing this new result in the revised manuscript.

*4) ATM is known to promote NF-κB signal, an important regulator of IL-8 expression. In this context, it is helpful to determine the impact of mutant p53 on NF-κB signal (e.g. phosphorylation status NEMO or levels and phosphorylation of p65), whether both ATM and mutant p53 promotes NF-κB signal to regulate IL-8 expression*.

Our data suggests that, like ATM, mutant p53 also regulates NF-κB signaling. We have shown that depletion of mutant p53 results in: 1) reduced IL-8 and IL-8 promoter activity (Figure 4), 2) reduced nuclear p65 (Figure 4) and 3) reduced p65 accumulation on the IL-8 promoter by ChIP analysis (Figure 4). These results were all obtained also in ATM depleted cells, suggesting that ATM and mutant p53 regulate this NF-κB pathway similarly. In response to the reviewers’ suggestions, we have performed additional experiments to analyze how mutant p53 could regulate NF-κB signaling. We observe that ATM or mutant p53-depleted cells do not reduce expression of NF-κB or NEMO at either the mRNA or protein levels (Figure 4—figure supplement 2). However, we did observe reduced phosphorylation of NEMO in p53-depleted cells that was similar to the reduction in ATM-depleted cells (Figure 4—figure supplement 2). These data, along with our new data showing the reduction of IL-8 mRNA in NEMO-depleted cells, suggest that NF-κB activity is regulated by ATM and mutant p53 through NEMO. Collectively, these data indicate that mutant p53 regulates NF-κB signaling in a similar fashion as ATM. These results are in agreement with ATM and mutant p53 cooperating to regulate NF-κB signaling that results in IL-8 expression.